# Recent Evidences of Epigenetic Alterations in Chronic Obstructive Pulmonary Disease (COPD): A Systematic Review

**DOI:** 10.3390/ijms26062571

**Published:** 2025-03-13

**Authors:** Rosetta Ragusa, Pasquale Bufano, Alessandro Tognetti, Marco Laurino, Chiara Caselli

**Affiliations:** 1Institute of Clinical Physiology, CNR, 56124 Pisa, Italy; pasqualebufano@cnr.it (P.B.); marco.laurino@cnr.it (M.L.); 2Department of Surgical, Medical and Molecular Pathology and Critical Care Medicine, University of Pisa, 56124 Pisa, Italy; 3Research Center E. Piaggio, University of Pisa, 56122 Pisa, Italy; alessandro.tognetti@unipi.it; 4Fondazione Toscana Gabriele Monasterio, 56124 Pisa, Italy

**Keywords:** COPD, epigenetic mechanisms, DNA methylation, histone modification, non-coding RNA

## Abstract

Chronic obstructive pulmonary disease (COPD) is a heterogeneous inflammatory condition characterized by progressive airflow limitation, which may be caused by genetic and environmental factors. Furthermore, epigenetic mechanisms could provide valuable insights into the complex interactions between environment and genes and subsequent development of the disease. The aim of this study is to provide a systematic review of the latest knowledge on epigenetic modifications that characterize COPD, summarizing epigenetic factors that could serve as potential novel biomarkers and therapeutic targets for the treatment of COPD patients. We queried the PubMed and Scopus electronic databases with specific keywords, in May 2024, according to the PRISMA guidelines, and articles were included if they met all the inclusion criteria and survived a quality assessment. We identified 5414 publications in our systematic search. Among them, only 51 articles met the criteria of COPD-associated epigenetic modifications in human patients compared to the control group. Eight studies described DNA methylation, one study histone modifications, and forty-two studies non-coding RNAs. Apoptosis and inflammatory pathways have been found to be the main mechanisms regulated by epigenetic elements in COPD patients. In addition, non-coding RNAs may be useful as biomarkers or therapeutic targets of pulmonary disease. Future studies will be needed to confirm the role of epigenetic elements associated with COPD.

## 1. Introduction

Chronic obstructive pulmonary disease (COPD) is a heterogeneous inflammatory condition that results in largely irreversible and progressive airflow limitation [1]. Before the COVID-19 pandemic, COPD was described as the third-leading cause of death worldwide, with approximately 3.3 million deaths, between 1990 and 2019 [2]. The Global Initiative for Chronic Obstructive Lung Disease (GOLD) report 2023 defined COPD as a “heterogeneous lung condition characterized by chronic respiratory symptoms (dyspnea, cough, expectoration and/or exacerbations) due to abnormalities of the airways (bronchitis, bronchiolitis) and/or alveoli (emphysema) that cause persistent, often progressive, airflow obstruction” and established the criteria for categorizing airflow limitation into four stages (GOLD classification): mild (GOLD1), moderate (GOLD2), severe (GOLD3), and very severe (GOLD4) COPD [3].

Cigarette smoking has long been considered the main risk factor for COPD [4]. The complex mixture found in tobacco smoke, which includes liquid droplets, volatile and semi-volatile compounds, and gases, can induce DNA damage, hinder the inflammatory response, and increase the production of reactive oxidative substances, leading to various levels of lung impairment [5]. However, the development of COPD even in non-smoking patients suggests that other elements besides environmental factors (cigarette smoking and air pollution) can contribute to the onset of disease. The genetic contribution to COPD is still being studied. The first genome-wide association study (GWAS) provided evidence of the association of the hedgehog interacting protein (HHIP), SERPINA, family with sequence similarity 13 member A (FAM13A), advanced glycosylation end-product specific receptor (AGER), cholinergic receptor nicotinic alpha 5 subunit (CHRNA5), and interleukin 27 (IL27) with COPD [6,7]. Nevertheless, GWAS studies conducted on larger samples have not confirmed these results, finding other gene variants associated with COPD, including genomic regions close to Ras and Rab interactor 3 (RIN3), cytochrome P450 family 2 subfamily A member 6 (CYP2A6), and desmoplakin (DSP) [6,8]. To date, GWAS analysis has identified COPD genetic loci that could only explain ~10% of European disease cases [7,8]. Furthermore, smoking and genetic variants considered as individual risk factors for lung disease have shown a moderate impact on COPD. Therefore, it has been suggested that COPD could be recognized as a progressive disease resulting from a combination of environmental, genetic, and epigenetic factors (Figure 1) [4].

Epigenetic regulation can occur at various levels, including DNA methylation, histone modifications, and modulation of non-coding RNA (ncRNA) [9,10]. Emerging evidence indicates that epigenetic regulation plays a crucial role in COPD development and progression (Figure 2) [9,10,11], and the reversibility of epigenetic alterations makes them valuable therapeutic targets for treating COPD patients [9,10]. Furthermore, the possibility of associating epigenetic alterations with disease progression could suggest the use of epigenetic elements as diagnostic and prognostic COPD biomarkers. This study aims to provide an objective, comprehensive, and updated overview of the most recent insights into the epigenetic modifications post-COVID-19 associated with COPD, and attempts to summarize epigenetic factors that could serve as potential new biomarkers and therapeutic targets for the treatment of patients with COPD.

## 2. Materials and Methods

### 2.1. Study Design and Search Strategy

We conducted this systematic review following the Preferred Reporting Items for Systematic Reviews and Meta-Analyses (PRISMA) guidelines [12,13]. PRISMA comprises a 27-item checklist to ensure and promote the quality of systematic reviews; this checklist is reported in Appendix A (for abstract checklist). The protocol employed in the current systematic review has been submitted for registration to the Open Science Framework platform (available online at https://osf.io/4ja25/; accessed on 31 January 2025). The retrieval process consisted of three phases.

#### 2.1.1. Systematic Search Phases

1. Preliminary research and definition of keywords

During the first phase, started in May 2024, we carried out a preliminary analysis of the literature in the absence of specific keywords to define the scientific databases to be analyzed, the keywords to be used, and the inclusion/exclusion criteria to be applied. Based on the research question and the exploratory research of literature, the strategy was to define keywords referring to the four macro areas of interest: (1) keywords related to COPD; (2) keywords related to molecules can be used to establish the presence, development, or treatment of COPD; (3) keywords related to epigenetic mechanisms; and (4) keywords related to pathophysiological mechanisms underlying COPD. The keywords belonging to each category are summarized in Table 1.

2. Systematic search and definition of PICOS

The second phase, conducted in May 2024, consisted of a systematic search among titles, abstracts, and keywords of scientific papers, using the electronic databases PubMed and Scopus, based on the selected keywords properly combined through the Boolean operators “AND” and “OR”. We limited our search to articles, written in English, and published from 1 January 2020 to the date of the search (9 May 2024) to include only the most recent papers in our systematic review and to avoid the massive inclusion of papers without dated analysis techniques. The search strategy, including all keywords used and the number of studies found from each database, is analytically reported in Table 2.

The final selection of papers for inclusion was carried out according to the Population, Intervention, Comparison, Outcomes, and Study Design (PICOS) worksheet [12,13], summarized in Table 3. We defined the following PICOS criteria: Population, we chose “Studies in Humans” and “Studies including COPD patients” as the inclusion criteria and “In vitro and in vivo studies” and “Participants with other malignancies” as exclusion criteria because we aimed to avoid including humans with COPD but with any malignancies that might influence the study results, thus preventing the introduction of bias; in addition, we excluded all studies conducted exclusively in vivo and in vitro. For Interventions, we chose as inclusion criteria “Assessment of DNA methylation, Histone modification, and ncRNA expression” to include only work that studied these epigenetic mechanisms to the exclusion of everything else. For Comparisons, we choose as inclusion criterion “Control group (between or within design)” to have a comparison between two populations to identify any differences in epigenetic mechanisms between healthy and COPD groups or during the evolution of the disease within patient groups. For Results, we chose as inclusion criteria “provide an unbiased and comprehensive overview of the current knowledge on epigenetic modifications associated with COPD” and “summarize epigenetic modifications translated into clinical therapeutic interventions and biomarkers for COPD”, to have a complete picture of the impact of epigenetic mechanisms in the disease, as well as the possibility of using epigenetic elements for the management of patients with COPD. For Study Design, we chose “Review, Scoping Review, Narrative Review, Systematic Review, Meta-Analysis, Editorial, Book, Case Report, Conference Review, and Conference Paper” as exclusion criteria following the guidelines for carrying out a systematic review.

3. Application of PICOS Study Design Exclusion Criteria

The final phase consisted of a first step in which, following PICOS criteria related to the Study Design section, we excluded all reviews, both narrative and systematic, meta-analyses, and conference papers, to substantially reduce the number of included studies.

#### 2.1.2. Title and Abstract Selection

By reading the titles and abstracts, we excluded all studies that did not match with the research question.

#### 2.1.3. Full-Text Selection According to PICOS Criteria

Finally, we included in the systematic review only clinical studies that investigated the epigenetic mechanisms in COPD. The papers included were read thoroughly to obtain the data of our interest.

#### 2.1.4. Synthesis Method

The papers included were clustered according to epigenetic mechanisms involved in COPD (molecular and cellular processes, as biomarker or therapeutic target). Table 4, Table 5, Table 6, Table 7, Table 8 and Table 9 describe the extracted information, including the following: Study = name of first author et al., year; Country (Region) = where the study took place; Number of participants = sample size; Type of sample = biological sample employed; Gene affected = gene or group of genes whose expression can be “regulated” by epigenetic mechanisms; Epigenetic alteration = type of epigenetic alteration observed in the presence of disease; Activity in COPD = involvement of epigenetic elements in different molecular and cellular mechanisms associated with COPD; and Role of epigenetic mechanisms = epigenetic modifications that can be used to explain the pathophysiology of COPD or as biomarkers and therapeutic targets.

#### 2.1.5. Study Risk of Bias Assessment

We assessed the risk of bias for each study in Appendix A by compiling the AXIS tool [62].

## 3. Results

### 3.1. Flow Diagram

Figure 3 shows all the phases in the systematic review process.

The research carried out on PubMed and Scopus yielded 2509 and 2905 studies, respectively. These 5414 papers were merged in a non-redundant database and 3094 papers remained. Then, by eliminating all studies not related to our research question and all studies that did not match PICOS criteria related to Study Design (exclusion for all types of reviews, meta-analyses, and conference papers), the number of studies was reduced to 172. Finally, by applying all the PICOS criteria, we obtained 51 studies to be included in the systematic review.

### 3.2. Study Selection and Characteristics

Two independent reviewers (R.R. and P.B.) checked the pool of 3094 abstracts collected from PubMed and Scopus search engine outputs; any disagreement was discussed with C.C. as the arbiter. Titles and abstracts were screened, and 2922 were removed through a semi-automatic script created via MATLAB R2024a (The MathWorks, Inc., Natick, MA, USA), according to PICOS criteria related to Study Design (875 papers) and irrelevance to the research question (2047 papers). The remaining 172 papers were checked for eligibility according to the remaining PICOS criteria; all in vitro and in vivo studies and studies with participant with other malignancies (84), all studies without assessment of DNA methylation, histone modification, or ncRNA expression (19), all studies with no control group (13), and lastly, all studies that did not provide a comprehensive overview of current knowledge on the topic or did not have a summary of epigenetic changes on the topic (5) were excluded. Finally, 51 articles were included, as summarized in Table 4, Table 5, Table 6, Table 7, Table 8 and Table 9, which shows the main information of each, divided according to epigenetic mechanisms in COPD patients.

### 3.3. Synthesized Findings

In this subsection, for each study, we report the main findings grouped according to epigenetic mechanisms in COPD patients.

Our search criteria identified 51 recent studies focusing on epigenetic mechanisms: DNA methylation (8 studies), histone modifications/chromatin remodeling (1 study), circRNAs (7 studies), lncRNAs (8 studies), miRNAs (25 studies), and competing RNAs (3 studies). All studies were conducted using human samples such as lung tissue, bronchoalveolar lavage (BAL) fluid, and blood. Moreover, several studies also employed in vitro and in vivo models to explain the effects of the epigenetic changes identified in clinical samples and to verify their use as potential therapeutic targets.

### 3.4. DNA Methylation

DNA methylation, the most studied epigenetic mechanism overall, is able to inhibit gene expression by adding a methyl group to cytosine and by recruiting proteins involved in gene transcription repression (Table 4) and it affects mainly specific sites called CpG islands (Figure 2) [63]. The imbalance in DNA methylation status can disrupt lung development, accelerating cellular aging and interfering with the inflammatory response to injuries [16].

Kachroo and colleagues investigated the relationship between fetal DNA methylation perturbations in fetal lung tissue exposed to maternal smoking, and DNA methylation in COPD development, using an original network (pathways)-CpG approach that focused on a few gene clusters relevant in COPD [14,15]. Via genome-wide DNA methylation assay, fetal lung samples (n = 42 tissue exposed to maternal smoking and n = 36 unexposed) were compared with those from adult-lung tissue (n = 160, 114 with COPD) [14,15]. No significant differences were found among the genome-wide DNA methylation assay of fetal lung tissue exposed to maternal smoking compared to unexposed [14,15], and CpGs in fetal lung tissue exposed to maternal smoking overlapped with DNA methylation profile in adult COPD patients [15]. In particular, the co-methylation in “gene modules” involved in cell growth and development, such as Wnt, phosphatidylinositol 3-kinase (Pi3K)/serine/threonine kinase (AKT), mitogen-activated protein kinase (MAPK), and serine/threonine-protein kinase (Hippo), in fetal lung tissues exposed to maternal smoking matched with the same co-methylation perturbations found in adult COPD [15]. These studies, also using a “gene promoter analysis” approach to validate COPD-related DNA methylation changes associated with age [14,15], suggested that, in adult subjects, specific CpGs associated with disrupted fetal development could be used to predict the chronological age (epigenetic clock) of the individual and to identify those subjects at high risk for age-associated lung diseases [14,15].

Schwartz and colleagues examined the whole epigenomic landscape including gene promoters and enhancers to identify DNA methylation changes associated with genes and pathways essential in COPD pathophysiology and progression [16]. Parenchymal fibroblasts were isolated from lung tissues of ex-smoker control subjects (no COPD), patients with mild COPD (stage I, according to the GOLD classification), and established COPD patients (stages II-IV). Tagmentation-based whole-genome bisulfite sequencing (T-WGBS) and RNA sequencing were performed to identify the profile of differentially methylated regions (DMRs) and to evaluate gene expression changes in COPD patients and control groups [16]. DMR profiles and changes in expression of genes involved in cell proliferation, DNA, repair, and extracellular matrix organization were found in COPD I patients compared to the control [16]. Moreover, the changes in DNA methylation, considering the gene promoter and enhancer region, suggested that all of the epigenetic landscape should be considered in the analysis of molecular mechanisms associated with COPD pathophysiology [16].

Eriksson Ström and colleague conducted epigenome-wide association studies (EWAS) on BAL cells from subjects with COPD and control subjects (smoker and ex-smoker) to assess DNA methylation patterns and to verify a potential relationship between age and disease manifestation or progression [17]. The main cell type in BAL was the macrophage, and 1155 differential methylation positions (DMPs) were found in COPD patients [17]. Moreover, 638 of 1155 DMPs showed higher mean methylation in COPD samples compared to the control [17]. The most significant impact of COPD-DNA methylation alterations was observed in genes such as zinc finger DHHC-type palmitoyltransferase 14 (ZDHHC14), negative regulator of ubiquitin-like proteins 1 (NUB1), NLR family pyrin domain containing 3 (NLRP3), and zinc finger protein 322 (ZNF322) [17]. A strong correlation among DNA methylation and chronological age was found in COPD patients, but no association between epigenetic changes and progression of illness was observed. Interestingly, 38.7% of DMPs were in proximity to COPD-associated single-nucleotide polymorphisms (SNPs), helping to explain why some smokers are prone to develop COPD while others are not [17].

DNA methylation clocks evaluated in peripheral blood could be used as biomarkers to stratify patients with COPD according to their risk for lung complications. Cordero and colleagues calculated epigenetic age in COPD and control subject using six DNA methylation clocks [DNAmAge (pan-tissue), DNAmAgeHannum (blood-specific), DNAmAgeSkinBlood (fibroblast- and blood-specific), DNAmPhenoAge DNAmGrimAge, and DNAmTelomereLength (DNAmTL)] [10]. Blood and airway epithelial samples were tested and the DNAmGrimAge pattern showed a strong correlation in epigenetic age between the blood and the airways [10]. Moreover, DNAmGrimAge was suggested to have a role as a blood epigenetic age biomarker for assessing accelerated aging in the airways of individuals with COPD and in the control group [10].

The differential methylation of two CpG sites of Pi3K catalytic subunit delta (Pi3KCD) (cg03971555 and cg12033075) in blood collected from COPD patients and control group was analyzed [18]. The CpG sites were hypermethylated in COPD patients compared to the control and were significantly associated with COPD mortality. Although the physio-pathological significance of these changes in DNA methylation has not yet been clarified, the measured Pi3KCD hypermethylation could be used as a prognostic biomarker of COPD mortality [18].

The recent studies on changes in DNA methylation in COPD patients increased the understanding of the disease’s molecular mechanisms, suggesting potential treatment targets. Nuclear factor E2-related factor 2 (Nfr2), a key antioxidant element, regulates the expression of glutathione peroxidase, heme oxygenase-1, and NAD(P)H quinone oxidoreductase, protecting the cells from oxidative damage [19]. Zhang and colleagues found aberrant hypermethylation in CpG sites of the Nrf2 promoter both in lung tissue and primary bronchial epithelial cells obtained from COPD patients compared to the control group [19]. Nrf2 downregulation led to increased oxidative stress, cell death, and the progression of COPD [19]. As a result, targeting the methylation of CpG sites in the Nrf2 promoter was suggested as a potential treatment approach against COPD progression [19]. Another potential therapeutic target proposed for the treatment of COPD is the aryl hydrocarbon receptor repressor (AHRR), a repressor of aryl hydrocarbon receptor (AHR), which regulated the xenobiotic metabolism and the apoptotic processes [20,64]. The methylation of the AHRR (CpG sites) gene promoter was lower in whole blood and airway epithelial cells from COPD patients compared to the control, but only in active smokers [20]. The increased levels of AHRR could modify epithelial cell proliferation, dysregulate mitochondrial function, and reduce apoptotic processes upon cigarette smoke exposure, promoting the worsening of lung function and the progression of COPD [20]. Therefore, regulation of AHRR CpG methylation could be considered as a therapeutic strategy for the treatment of COPD [20].

### 3.5. Histone Modifications

Histone modifications are epigenetic mechanisms involved in regulation of gene transcription, genomic instability, premature aging, inflammatory activation, and DNA damage/repair (Figure 2) [65]. Several studies before 2020 had already suggested that histone modifications are associated with COPD presence and progression [21]. In the most recent study conducted in 2022, elevated protein arginine methyltransferase 7 (PRMT7) expression levels were detected in subjects with COPD compared to smoker controls [21]. In monocytes, PRMT7 is able to methylate histones on the arginine residues of the Ras-related protein 1A (Rap1a) enhancer, increasing the adhesion and migration capacity of monocytes on lung tissue, thus negatively affecting the outcome of COPD patients [21]. These data were confirmed by an in vivo study, in which Prmt7+/− mice exposed to cigarette smoke (CSE) showed less monocyte/macrophage infiltration in the lungs and were protected from chronic lung disease [21], suggesting the use of PRMT7 inhibitors as a possible therapeutic strategy to treat COPD.

### 3.6. Non-Coding RNA

#### 3.6.1. Circular RNA

Circular RNAs (circRNAs) are a type of non-coding RNA that form closed loops due to the lack of free 3′ and 5′ ends (Figure 2) (Table 6). Some circRNAs have multiple miRNA-binding sites and can prevent miRNA-mRNA targets from binding, a phenomenon known as the “sponge effect”. circRNAs can also act as protein sponges, protein decoys, and scaffolds for forming protein complexes. Furthermore, a few circRNAs have been found to regulate gene expression [66]. Aberrant expression of circRNAs is involved in the pathogenesis of several diseases including cardiovascular and pulmonary disorders, such as lung cancer, pulmonary fibrosis, tuberculosis, and acute lung injury [66]; however, the association between circRNA and COPD development is poorly explored. Liu and colleagues found a differential expression of circRNAs (N = 81) between COPD and healthy subjects [24], focusing on circ_FCHO2, circ_MBOAT2, circ_PTPN22, circ_TBC1D22A, circ_ACADM, and circ_CKAP5 [24]. Bioinformatics tools demonstrated that these six circRNAs, which act as the sponges of 18 miRNAs, are also differentially expressed between COPD patients and healthy subjects [24].

Two recent studies examined the expression profiles of circRNAs in peripheral blood mononuclear cells (PBMCs) of COPD patients and compared them with those of control subjects [22,23]. In the first study, 2132 circRNAs were found to be differentially expressed in patients with COPD compared to normal controls [22]. The mRNA, targeted from dysregulated circRNA obtained by gene ontology (GO) enrichment analysis, indicated that various signaling pathways, including cell death and apoptosis, as well as MAPK activation, may play a role in molecular and cellular processes associated with COPD development [22]. Moreover, Kyoto Encyclopedia of Genes and Genomes (KEGG) molecular pathway enrichment analysis revealed that circRNAs differentially expressed in COPD could influence disease development by affecting NOD-like receptor signaling, Toll-like receptor signaling, and Type 1 T helper (Th1)/Th2 cell differentiation [28]. In the second study, the analysis of the circRNA profile in PBMCs showed high expression levels of hsa-circ_0008833 in COPD patients compared to controls [23]. The functional investigation analysis suggested that circ_0008833 could promote COPD progression by inducing a distinct form of inflammatory cell death (pyroptosis) in bronchial epithelial cells [23]. Hsa-circ_0008833, derived from the SMAD3 gene, is capable of encoding a small peptide (57aa); an in vitro study using 16HBE bronchial epithelial cells confirmed the ability of the hsa-circ-0008833-57aa pathway to increase the mRNA of Caspase 1, IL-18, and IL-1β, inducing pyroptosis [23].

Zhang and colleagues examined the effect of circ_0006892 in bronchial epithelial cell injury [25]. Circ_0006892 acts as a sponge for miR-24, thus reducing the bronchial epithelial cell inflammatory injury [25], confirming that the circ_0006892 expression was downregulated in the lung tissue of COPD smokers compared to COPD non-smokers [25]. Moreover, in COPD patients, the disequilibrium in circ_0006892-miR-24 reduced the expression of anti-apoptotic and inflammatory elements, thus promoting worsening of COPD processes through low levels of PH domain and leucine-rich repeat protein phosphatase 2 (PHLPP2) [25]. Differently from circ_0006892, the expression of circ_ANKRDII was found to be upregulated in the lung tissue collected from COPD smoker patients when compared to the smoker and non-smoker subjects without COPD [26]. The authors demonstrated that circ_ANKRDII binds to miR-145-5p, neutralizing its protective effects and contributing to the molecular mechanisms underlying COPD pathogenesis, such as inflammation, oxidative stress, and apoptosis [26].

CircRNAs, owing to their distinctive structural conformation, which makes them particularly stable in the circulation, have been suggested as potential biomarkers. In a recent study, 119 circular RNAs were identified to be differentially expressed in plasma samples from individuals with very severe COPD compared to the control group (90 upregulated and 29 downregulated circular RNAs) [27]. After validation by real-time PCR, only the four circRNAs confirmed as differentially expressed between the two groups were considered as potential prognostic biomarkers of severe COPD [27]. Furthermore, GO and KEGG enrichment analysis indicated that circ_0008882, circ_00089763, circ_00062683, and circ_00077607 may be involved in the onset and progression of COPD by influencing immune cell development and activity [27]. Another study by Shen and colleagues suggested a role for circ_0049875 and circ_0042590 as diagnostic biomarkers of COPD [28]. They observed that circ_0042590 was upregulated while circ_0049875 was downregulated in PBMCs collected from COPD compared to controls, finding a strong correlation between circ_0049875 and circ_0042590 levels and acute exacerbation of COPD [28].

#### 3.6.2. Long Non-Coding RNA

Long non-coding RNAs (lncRNAs) are a type of non-coding transcripts longer than 200 nucleotides and characterized by secondary or tertiary structures that resemble proteins (Figure 2) (Table 7) [67]. Most lncRNAs are found in the nucleus, but can also be detected in cytoplasm, serum, and extracellular vesicles, and play a role in regulating gene expression through epigenetic, transcriptional, and post-transcriptional mechanisms [67].

Several studies before 2020 had already suggested that, among lncRNAs, lncTUG1 and ANRIL may be involved in inflammatory process, affecting the airway epithelial cells of COPD patients [68,69]. Studies conducted since 2020 have confirmed the close relationship between inflammation/lncRNAs/COPD. Most recently, Wu and colleagues explored the role of the lncRNA-IL7 receptor (lncRNA-IL7R) in COPD pathophysiology [29]; this is an anti-inflammatory lncRNA-s induced by toll-like receptor (TLR) 2 and 4 activation [29]. In PBMCs, lncRNA-IL7R is located in the nucleus and can regulate the expression of IL-8 by the deacetylation of lysine 9 on histone H3 (H3K9), and the tri-methylation of lysine 9 on histone H3 (H3K9me3) and histone H3 lysine 27 (H3K27me3) [29]. Low expression levels of lncRNA-IL7R have been correlated with inflammation and acute exacerbation phenotype of COPD [29].

Long non-coding RNA homeobox A cluster antisense RNA 2 (lncRNA- HOXA-AS2) is located between the human HOXA3 and HOXA4 gene [38]. The role of lncRNA- HOXA-AS2 in cellular and molecular mechanisms associated with COPD was explained by an in vitro study using human pulmonary microvascular endothelial cells (HPMECs) under CSE [30]. Low levels of lncRNA- HOXA-AS2 in HPMECs have been linked to reduced cell proliferation and viability, contributing to endothelial dysfunction through the activation of inflammatory responses [30]. LncRNA- HOXA-AS2 was downregulated in lung tissue samples collected from COPD patients compared with controls and its effects were mediated by notch receptor 1 (Notch1) (a receptor capable of monitoring proliferation, differentiation, apoptosis, and even stem cell maintenance) [30].

Besides their role in the pathogenesis of chronic pulmonary disease, lncRNAs are also proposed as novel biomarkers for COPD exacerbation. Wang and colleagues investigated the ability of lncRNA-plasmacytoma variant traslocation 1(lncRNA-PVT1) as a predictor of COPD exacerbation [31]. The expression levels of lncRNA-PVT1 were evaluated in PBMCs collected from healthy subjects, stable COPD, and acute exacerbation COPD (AECOPD) patients [31]. Data indicated that lncRNA-PVT1 levels were significantly elevated in COPD patients compared to controls and could distinguish AECOPD patients from stable COPD [31]. Previous research indicated that lncRNA-PVT1 plays a pro-inflammatory role in other diseases by reducing the function of miR-146a [70,71]. LncRNA-PVT1 levels were positively correlated with expression of inflammatory mediators such as IL-6, TNFα, IL-8, and IL-17 in both stable COPD and AECOPD patients [31]. Furthermore, the expression of anti-inflammatory miR-146a showed a negative correlation with lncRNA-PVT1 levels and the progression of COPD [31]. These findings suggest that the pro-inflammatory pathway of lncRNA-PVT1 relied on the inhibition of its specific target, miR-146a [31]. Similarly, lncRNA metastasis-associated lung adenocarcinoma transcript1 (lncRNA-MALT1) has also been proposed as a potential biomarker for predicting the risk of acute exacerbation and disease progression in COPD [32]. Circulating lncRNA-MALAT1 levels were positively associated with GOLD stage and inflammatory mediator (TNFα, IL-6, IL-8, IL-1β, IL-17 and IL-23) levels and negatively related to expression levels of miRNA potentially involved in the reduction in inflammatory processes (miR-133, miR-125b, miR-146a, and miR-203) [32]. Moreover, blood levels of lncRNA-MALAT1 were described as a good way of distinguishing AECOPD from COPD patients [32]. A recent study proposed lncRNA-cancer susceptibility candidate 2 (lncRNA-CASC2) as a diagnostic biomarker of COPD [33]. In an in vitro study using a human bronchial epithelial cell line, it was observed that under normal conditions, lncRNA-CASC2 acts as a sponge for miR-18a-5p, reducing the expression of pro-inflammatory cytokines [33]. Circulating levels of lncRNA-CASC2 were found to be significantly decreased in COPD patients compared to smoking control subjects and were able to distinguish severe COPD from mild and moderate cases [33]. Moreover, the levels of lncRNA-CASC2 were negatively correlated with increased miR-18a-5p and inflammatory cytokines, thus confirming its role in regulating inflammation [33].

High circulating levels of lncRNA-LUCAT1 were detected in the blood samples collected from patients with liver, pulmonary, gastric, and breast cancers [44]. Zhao and colleagues investigated the expression of lncRNA-LUCAT1 in COPD and its potential role as biomarker [34]. The data indicated that serum levels of lncRNA-LUCAT1 could effectively distinguish COPD patients from healthy individuals and smokers with COPD compared to non-smokers with COPD [44]. Mechanistically, it was suggested that lncRNA-LUCAT1 impaired the anti-inflammatory and anti-apoptotic effects of miR-181-5p in bronchial epithelial cells, thereby contributing to the progression of COPD [34].

Recent studies have highlighted the possibility of also using lncRNAs as therapeutic targets for COPD [35,36]. The lncRNA-HOX transcript antisense (lncRNA-HOTAIR) is involved in various carcinogenic processes, and, specifically, was upregulated in COPD lung tissue compared to healthy tissue [35]. In human pulmonary endothelial cells treated with CSE, the level of lncRNA-HOTAIR increased, thus contributing to the apoptosis process by hypermethylation of the Bcl-2 gene promoter [35] and to the increased levels of IL-1β and IL-6. The expression levels of lncRNA-colon cancer-associated transcript 1 (lncRNA-CCT1), similarly to lncRNA-HOTAIR, was found to be upregulated in lung tissue collected from COPD patients compared to the control group [36]. Mechanistically, it was suggested that lncRNA-CCT1, through the inhibition of the anti-inflammatory miR-152-3p, could promote the inflammatory processes by activating ERK signaling [36]. Given the role of lncRNA-HOTAIR and lncRNA-CCT1 in the molecular mechanisms linked to COPD, these findings indicate that these lncRNAs may serve as novel therapeutic targets for COPD prevention [35,36].

#### 3.6.3. MicroRNA

MicroRNAs (miRNAs), a family of single-stranded, non-coding small RNAs, approximately 22 nucleotides in length (Table 8), play a crucial role in regulating various physiological and pathological processes by interacting in a sequence-specific manner with mRNAs in the cytoplasm (Figure 2) [72]. miRNAs directly influence gene expression by recruiting different protein complexes to the promoter or enhancer regions of target genes, which can lead to either the activation or repression of transcription [72]. Several studies explored the role of abnormal miRNA expression in COPD patients, indicating their significant function in the pathophysiology of the disease and their potential use as biomarkers and therapeutic targets for COPD.

The analysis of PBMCs’ transcriptome and miRNome profiles in severe COPD patients and a control group was assessed to identify the mRNAs and miRNAs associated with disease [37]. The application of bioinformatics tools to identify altered pathways in PBMCs from COPD patients suggested that IL-8 and inducible T cell costimulator (iCOS)-iCOS ligand (iCOSL) signaling were primarily involved in disease progression [37]. Among differentially expressed miRNAs, 7 miRNAs were found to be able to target genes involved in IL-8 signaling, while 12 miRNAs were associated with iCOS-iCOSL signaling regulation, suggesting a role for these miRNAs in inflammatory processes [37]. In addition to PBMCs, miRNome was also carried out on BALF and plasma samples collected from COPD patients and compared with a control group [38]. A total of six miRNAs-BALF were found differentially expressed between the groups, while only three miRNAs were significantly downregulated in COPD plasma samples compared to the control [38]. The GO enrichment analysis and KEGG analysis indicated that the differentially expressed miRNAs in both BALF and blood contributed to the oxidative and inflammatory processes underlying COPD through the regulation of the MAPK, RAS, and FOXO signaling pathways [38].

In addition to exploratory studies, other research has examined the specific role of miRNAs in COPD. The impact of miR-486-5p on COPD was studied using alveolar macrophages and peripheral monocytes; results showed that its expression levels were significantly upregulated in monocytes/macrophages collected from COPD patients compared with control group [50]. Moreover, the increased levels of miR-486-5p were associated with TLR4 and pro-inflammatory cytokine expression [39]. An in vitro study using a CSE/lung alveolar macrophage cell line confirmed that miR-486-5p contributed to the overexpression of TLR4 and cytokines by inhibiting the expression of histone acetyltransferase (HAT1) [39]. Moreover, high circulating levels of miR-486-5p together with miR-106b-5p were observed in blood samples from COPD patients when compared to the control group, suggesting the possible use of these ncRNAs as diagnostic biomarkers of hypoxia and pulmonary hypertension (PH) [49].

Still in the context of inflammation in COPD, contrasting results were obtained studying the expression levels of miR-221-3p in COPD patients [40,50]. The miR-221-3p expression was downregulated in COPD lung tissue compared to controls [40], and the low levels of miR-221-3p were negatively related with expression of cyclin dependent kinase inhibitor 1B (CDKN1B), pro-inflammatory, and apoptotic elements [40]. In contrast to the lung expression, in the serum samples collected from AECOPD and stable COPD patients, the circulating levels of miR-221-3p were significantly higher compared to controls [50]. CSE-treated 16HBEC expressed higher miR-221-3p levels compared to untreated cells. Moreover, the increased levels of miR-221-3p were positively correlated with the mRNA of IL-6, IL-1β, TNFα, Collagen IV, Fibronectin, and α-SMA [50]. Finally, Shen and colleagues suggested that circulating miR-221-3p in combination with miR-92a-3p could be used as a diagnostic biomarker of COPD in order to distinguish COPD exacerbation from stable COPD [50].

The expression of Receptor for Advanced Glycation End-products (RAGE), a multi-ligand receptor of the immunoglobulin superfamily able to activate ROS signaling and the MAPK pathway, was higher in lung tissues collected from COPD patients compared to controls [41]. In vivo and in vitro studies showed that increasing miR-23a-5p levels could reduce the inflammatory and oxidative processes involved in COPD, specifically by targeting RAGE [41], and its anti-inflammatory effect might be absent in COPD, considering the low expression levels of miRNA in patients compared with the control group [41]. Other inflammatory miRNAs linked to the pathogenesis of COPD are miR-21 and miR-155 [42,43]. Their expression levels were found to be upregulated in bronchial biopsies taken from COPD patients compared to healthy subjects [42,43]. By inhibition of the expression of the anti-inflammatory factor special AT-rich binding protein 1 (SATB1), miR-21 was observed to be able to induce an increased expression of chemotactic factors, such as calgranulins S100 calcium binding proteinA9 (S100A9), and the activation of the “inflammatory transcription factor” NF-kB [55]. In patients with COPD, elevated lung levels of miR-155 could contribute to inflammation by inducing the expression of MMP12 and ADAM19, which are proteases that cleave extracellular matrix proteins and release cytokines, such as TNFα [43].

Zhu and colleagues investigated the role of miR-103a in COPD pathophysiology and discovered a connection between miRNA and the impaired function of alveolar macrophages in COPD patients [44]. Surfactant proteins are produced from Type II alveolar cells to preserve surface tension, and CSE increased the oxidation of surfactant proteins. Under normal conditions, alveolar macrophages should remove oxidized proteins, protecting alveolar compliance; however, in alveolar COPD patients, the increase in oxidized proteins was associated with the formation of lipid-laden macrophages or foam cells [44]. Therefore, the downregulation of miR-103a in alveolar macrophages, collected by bronco alveolar lavage from COPD patients, could explain the dysfunction of these inflammatory cells [44]. Both in vitro and in vivo studies confirmed that CSE induces a reduction in the expression of miR-103a by increasing surfactant oxidative protein, which leads to an increase in its mRNA target LDLR and results in lipid and oxidized lipid accumulation in macrophages [44].

One of the most common complications of COPD is PH, characterized by the remodeling of the pulmonary vessels, leading to their thickening and stiffening [73]. Yang and colleagues explored the miRNAs potentially involved in COPD-PH severity and progression [45]. miR-4640-5p was upregulated in lung tissues collected from COPD-PH compared to a normal lung [45] and high levels of miR-4640-5p were able, in pulmonary vascular smooth muscle cells, to induce a decrease in NOS1 expression and an increase in cell proliferation [45], thus causing thickening of the vessel wall and progression of the disease [45]. In addition to PH, higher production and accumulation of mucus in airways was observed in COPD patients; in fact, Singh and colleagues verified that in the lung tissue of COPD smokers, the expression levels of Anoctamin 1 (ANO1), a calcium-activated chloride channel present in the airway epithelium, involved in mucus secretion and regulation of vascular contraction, were higher compared to the control [47]. Moreover, the exposure to low doses of cadmium, a toxic substance released by burning tobacco, increased the expression of ANO1 by downregulating miR-381 [47]. In vitro studies confirmed that miR-381 directly regulates ANO1 mRNA, suggesting that miR-381 could be a potential target of treatment for patients with COPD [47]. In addition to mucus secretion, in COPD patients, the cadmium exposition and accumulation in the blood can contribute to epithelial-mesenchymal transition (EMT) and COPD progression [48]. Elevated levels of cadmium in plasma correlated with a decrease in the pulmonary expression of epithelial markers, such as E-cadherin, and with an increase in mesenchymal biomarkers, such as N-cadherin, Vimentin, and α-SMA [48]. Zheng and colleagues found that circulating levels of miR-30 negatively correlated with plasma cadmium concentration in COPD patients [48]. Furthermore, in vitro experiments confirmed that exposure to cadmium promoted EMT in lung epithelial cells by reducing miR-30 expression [48].

Other miRNAs associated with chronic mucus hypersecretion (CMH) in COPD patients were observed [74]. Tasena and colleagues investigated the role of some CMH-associated miRNAs in aberrant fibroblast–epithelial cell crosstalk in COPD patients [46]. Using a co-culture of primary bronchial epithelial cells (PBECs) and pulmonary airway fibroblasts (PAFs) collected from COPD-CMH patients, an increase in the expression levels of miR-708-5p was observed in the PBECs. In contrast, let-7a-5p and miR-31-5p showed higher expression levels in the PAFs after their interaction with the epithelial cells [46]. Tasena and colleagues observed that in physiological conditions, the expression levels of miR-708-5p decreased in PBECs during mucociliary differentiation and that an increase in miR-708-5p suppressed MUCS5AC secretion, as demonstrated after miRNA-mimic transfection study [46]. Considering that expression of miR-708-5p is increased in PBECs collected from COPD patients, it was suggested that the regulatory mechanisms of this miRNA on MUCS5AC were altered by other factors [46]. The changes in the expression levels of let-7a-5p and miR-31-5p in co-cultured PAFs were linked to a decrease in the expression of COL4A1 and COL5A1. This decrease may contribute to abnormalities in the basement membrane, which could in turn affect epithelial differentiation and promote CMH [46].

The necessity of identifying diagnostic and prognostic biomarkers for the early detection of COPD onset and progression has prompted numerous studies to explore the potential use of miRNAs as markers in patients suffering from chronic lung disease. In this context, several studies reported that circulating miR-548ar-3p, miR-146a, miR-218, miR-150-5p, miR-423-5p, miR-1290, and miR-1246 were under-expressed in COPD patients compared to the control group [51,52,53,54,55]. miR-146a and miR-218, involved in the inflammatory response and oxidative processes, were able to discriminate COPD smokers from COPD non-smokers [52], while miR-150-5p, miR-423-5p, miR-1290, and miR-1246 could be used as markers of worsening disease, considering that the circulating levels of miRNAs were relatively lower in patients with severe decline in lung function than in those with stable COPD [53,54,55]. However, Cazola-Rivero and colleagues studied the changes in the circulating levels of miR-1246 during 10 years of follow-up in relation with COPD progression, suggesting that circulating levels of miR-1246 do not effectively differentiate between COPD patients and healthy individuals [56]. Furthermore, no significant changes in miR-1246 levels were observed in COPD patients at follow-up compared to baseline measurements [69]. Finally, it was confirmed that miR-1246 could distinguish between COPD patients and those with both COPD and emphysema, suggesting the use of miRNA as a biomarker of disease severity [56].

Conversely, circulating levels of miR-4433a-5p, miR-126, and miR-210 were elevated in COPD patients compared to controls [55,57,58]. The expression of miR-4433a-5p gradually increased in blood samples of COPD patients with the progression of disease [55]; in fact, miR-4433a-5p targeting the PIK3R2 gene could promote the apoptosis and inflammatory processes in COPD patients [55]. Wang and colleagues found that circulating levels of miR-126, a marker of inflammation that can contribute directly to processes regulating the synthesis of pro-inflammatory cytokines, were elevated in COPD patients compared to a control group. This miRNA was also able to differentiate patients with acute exacerbations of the disease from those with stable COPD [57]. Finally, miR-210 was described as an early biomarker of PH in COPD patients [58]; however, no differences in blood miR-210 levels were found when stable COPD patients were compared with healthy subjects [58]. Instead, in patients with COPD-induced pulmonary hypertension, miR-210 expression was significantly higher compared to patients with stable COPD [58].

In addition to free miRNAs in various biological fluids, it is possible to identify miRNAs encapsulated within extracellular vesicles (EVs), which could also be used as biomarkers. Very recently, the profile of EV-associated miRNAs in BALF was compared between COPD patients and a control group [59]. The number of EVs in the control groups was lower than in COPD patients, and the combination of EVs miR-2110, miR-223-3p, and miR-182-5p was able to distinguish between healthy patients and patients with lung disease [59]. In addition, miR-2110 and miR-182-5p were associated with neutrophil expression [59] and, in contrast, EVs-miRNAs like miR-223-3p, miR-338-3p, and miR-204-5p were linked to eosinophil expression. This suggests that these specific EVs-miRNAs could be used to identify the inflammatory patterns underlying COPD [59]. Among the EVs, exosomes, small nanovesicles released into the extracellular space, were studied as carriers of miRNAs in several diseases. In serum, the levels of free or exosom-miR-1258 were found to be significantly higher in COPD patients compared to controls [60]. Moreover, exosome-miR-1258 proved to be a more accurate marker for distinguishing COPD patients experiencing acute exacerbations of the disease from those with stable COPD [60].

In a recent work, miRNAs were proposed as therapeutic targets in treating patients with COPD [61]. It is well known that chronic inflammation disrupts the balance between apoptosis and cellular proliferation, leading to one of the adverse effects of COPD progression: airway endothelial cell hyperplasia [61]. miR-196-5p and miR-361-5p were able to promote hyperplasia in endothelial cells by reducing mRNA-ARHGEF12 (miR-196-5p) and increasing the expression levels of mRNA-BCAT1 (miR-361-5p) [61]; they were differentially expressed in blood samples collected from COPD patients compared to controls [61]. In an in vitro study, gamma-sterol (GS) was shown to reduce epithelial cell hyperplasia by decreasing the expression of miR-196-5p and increasing the levels of miR-361-5p, suggesting its possible use in the treatment of COPD patients [61].

#### 3.6.4. Competing Endogenous RNA

In recent years, a new theory called “competing endogenous RNA” (ceRNA) has been explored to study various diseases [75]. The ceRNA theory suggests that mRNA, lncRNA, and circRNA could act as miRNA sponges, thereby regulating the expression of specific mRNAs and biological networks [75]. In relation to the development of COPD, two works have attempted to identify the ceRNA networks (Table 9) [9,11]. The mRNA/lncRNA/circRNA/miRNA profiles were analyzed in lung tissue collected from COPD patients and healthy subjects [9,11], and 1289 mRNAs, 69 miRNAs, 32 circRNAs, and 433 lncRNAs were found to be differentially expressed between the two groups [9,11]. After an appropriate bioinformatics analysis, the authors concluded that these ceRNAs differentially expressed between healthy and COPD subjects could be involved in the regulation of 18 gene hubs, including TGFβ signaling and Wnt/β-catenin signaling, immune cell infiltration, M2 macrophage differentiation, and NK cell activation [9,11].

## 4. Discussion

This systematic review summarized the available evidence regarding epigenetic alterations in COPD patients from studies published after the pandemic emergency due to SARS-CoV-2. The current evidence provides a detailed description of epigenetic elements, and their related pathways involved in COPD onset, which could be used for the development of the diagnostic and therapeutic strategies for the lung disease. A key aspect highlighted in many publications is that, at the molecular level, the epigenetic elements associated with COPD play a role in regulating the expression of factors involved in inflammation, oxidative stress, apoptosis, and cell proliferation. Some studies showed that COPD patients exhibited elevated levels of circ_0008833, miR-486-5p, and miR-21, while the levels of lncR-17R were decreased in PBMCs and/or lung tissue when compared to a control group [23,29,39,42]. These changes contributed to increased expression of pro-inflammatory cytokines and chemokines and to impaired lung function. On the other hand, aberrant expression levels of circANKRDII, lncR-HOXA-AS2, miR-221-3p, and miR-23a-5p resulted in dysregulation of apoptosis/cell proliferation and oxidative stress [26,30,40,41]. Moreover, some miRNAs were also able to interfere with processes essential for disease progression, such as mucus production, and with mechanisms related to both epithelial–mesenchymal transition and PH [45,46,47].

Epigenetic studies have also highlighted great potential to identify probable biomarkers of COPD, and 17 out of the 51 studies included in this systematic review discuss the use of epigenetic factors as potential biomarkers. They highlighted a lengthy list of circRNAs, lncRNAs, and miRNAs related to an inflammatory process as being able to indicate the presence of a disease. Considering the studies involving large samples of COPD patients compared with control groups, ncRNAs, including circ_00089763, circ_0008882, lncR-PVt1, lncR-MALAT1, miR-126, miR-221-3p, and miR-92a-3p, have been described as excellent inflammatory biomarkers that can establish the presence of COPD [27,31,32,50,57]. Studies on changes in DNA methylation related to COPD reported that alterations in CpG methylation linked to disrupted fetal development are similar to DNA methylation patterns observed in adult COPD patients [14,15]. These specific DNA methylation regions could potentially help in predicting which individuals are at a higher risk for developing age-related lung diseases.

The most interesting aspect of epigenetic modifications is their reversibility, which places them as promising candidates for pharmacological treatments. This review reports the most recent studies on treatments of COPD that target epigenetic factors. Among them, Chen and colleagues suggest that restoring AHRR gene methylation (which reduces AHRR mRNA and protein levels) may be an effective therapeutic strategy. This approach may improve mitochondrial function and help reduce programmed cell death in COPD patients exposed to cigarette smoke [20]. Another promising therapeutic target for preventing COPD has been identified in Lnc-HOTAIR; in fact, a reduction in its expression may lead to decreased apoptosis by enhancing Bcl-2 production in the lung [35]. These studies will require further verification, also considering that the dysregulation of multiple epigenetic factors occurs during the onset of COPD, and thus re-modulating only the methylation of a gene or the expression of a ncRNA may not have the desired therapeutic effect.

This systematic review emphasizes that the main critical issue in epigenetic studies of COPD is the variability in sample size, type, and methods used for evaluating epigenetic changes. Furthermore, while most of the studies on DNA methylation and histone modification were conducted in European and American populations, the studies on non-coding RNA were carried out in China.

Finally, the above data show that epigenetic regulation affects the pathophysiology of COPD disorders with potential diagnostic or therapeutic utility. Factors like ethnicity and environmental influences may act as confounding variables in the study of epigenetic mechanisms related to COPD. Therefore, it is essential to approach the topic of epigenetics in COPD by considering multiethnic studies. This approach could help to reduce these confounding factors and reveal the fundamental mechanisms that regulate the onset of the disease.

## Figures and Tables

**Figure 1 ijms-26-02571-f001:**
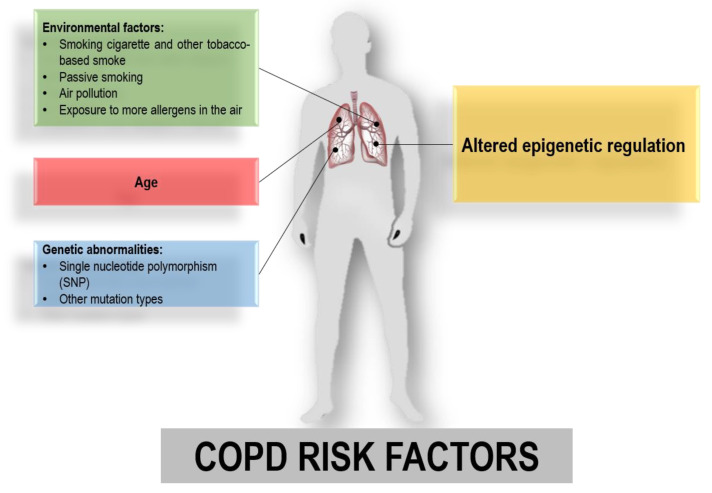
Risk factors associated with chronic obstructive pulmonary disease (COPD): environmental factors (green box) and age (red box), genetic abnormalities (blue box), and alternated epigenetic modifications (yellow box).

**Figure 2 ijms-26-02571-f002:**
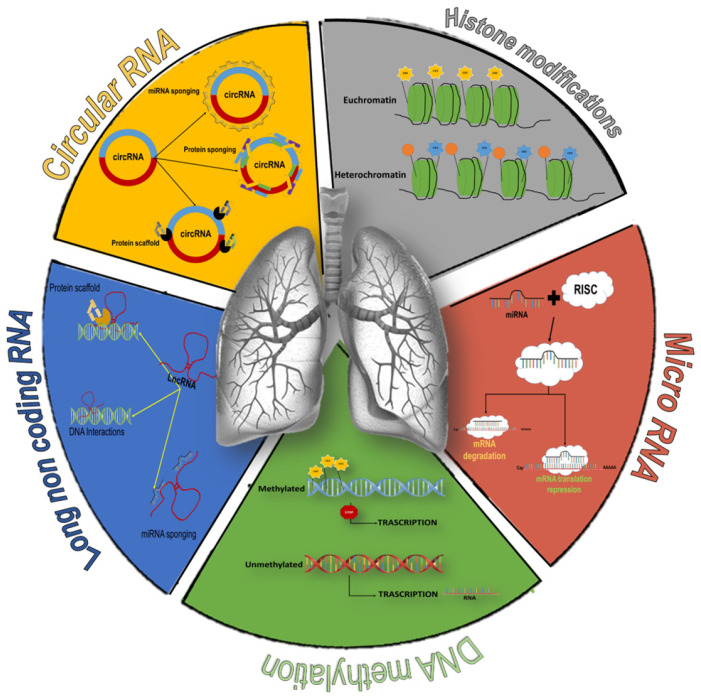
Overview of epigenetic modifications associated with COPD. DNA methylation: addition of a methyl group to cytosine, promoting the de-acetylation of histones and the heterochromatin state. The removal of methyl groups favors the formation of less compact chromatin (euchromatin) and favors transcription (green element). Histone modifications: acetylation of histone tails allow chromatin to assume a less condensed conformation permissive for transcription, while deacetylation of histone protein increases chromatin packaging, thus preventing DNA transcription (grey element). Circular RNA: sponges for miRNAs or proteins and scaffolds for the formation of protein complexes (orange element). Long non-coding RNA: role in regulating gene expression through transcriptional mechanisms (generating hybrid structures with DNA or recruiting transcription factors) or post-transcriptional mechanisms (miRNAs sponging) (blue element). microRNA: regulation of gene expression by acting as a guide strand that recognizes target mRNAs, inhibits their translation (complete miRNA-mRNA pairing), or promotes destabilization of target mRNAs (red element).

**Figure 3 ijms-26-02571-f003:**
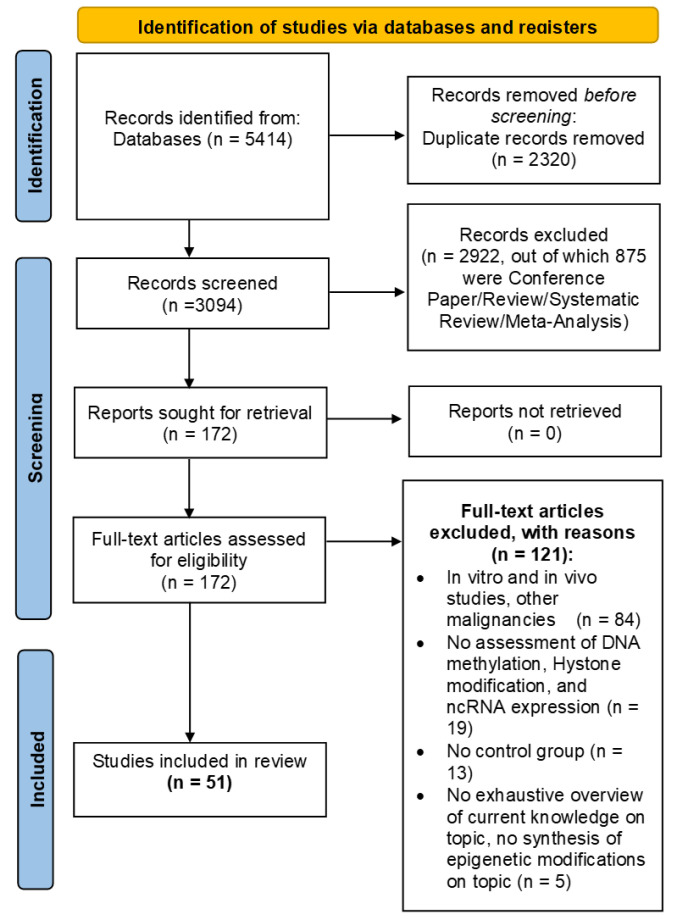
Flow Diagram.

**Table 1 ijms-26-02571-t001:** Keywords.

Key Words
Chronic obstructive pulmonary disease, COPD

Biomarker*, therapeutic target*

Epigenome-Wide Association, sequencing, epigenetic*, DNA methylation*, long noncoding RNA*, circRNA*, miRNA*, histone* deacetylation*, histone* protein*, HDAC
Molecular mechanism*, extracellular vesicle*

**Table 2 ijms-26-02571-t002:** Search strategy.

Database	Steps	Query	Research in	Items Found
**PubMed**	#1	(((((((((((((Biomarker*[Title/Abstract]) OR (“therapeutic target*”[Title/Abstract])) OR (“Epigenome-Wide Association*”[Title/Abstract])) OR (sequencing[Title/Abstract])) OR (epigenetic*[Title/Abstract])) OR (“DNA methylation*”[Title/Abstract])) OR (“long noncoding RNA*”[Title/Abstract])) OR (circRNA*[Title/Abstract])) OR (miRNA*[Title/Abstract])) OR (“histone* deacetylation*”[Title/Abstract])) OR (“histone* protein*”[Title/Abstract])) OR (HDAC[Title/Abstract])) OR (“Molecular mechanism*”[Title/Abstract])) OR (“extracellular vesicle*”[Title/Abstract])	Title/Abstract	1,460,188
	#2	“COPD”[Title/Abstract] OR “Chronic obstructive pulmonary disease”[Title/Abstract]	Title/Abstract	83,977
	#3	Combine #1 AND #2		5834
	#4	Limit to (English)		5663
	#5	Limit after 2020		2509
**Scopus**	#1	TITLE-ABS-KEY (“Biomarker*” OR “therapeutic target*” OR “Epigenome-Wide Association*” OR “sequencing” OR “epigenetic*” OR “ DNA methylation*” OR “long noncoding RNA*” OR “circRNA*” OR “miRNA*” OR “histone* deacetylation*” OR “histone* protein*” OR “HDAC” OR “molecular mechanism*” OR “extracellular vesicle*”)	Title/Abstract/Keywords	1,989,075
	#2	TITLE-ABS-KEY (“COPD” OR “Chronic Obstructive Pulmonary Disease”)	Title/Abstract/Keywords	102,296
	#3	Combine #1 AND #2		7433
	#4	Limit to (English) and (Italian)		7078
	#5	Limit after 2020		2906

**Table 3 ijms-26-02571-t003:** Population, intervention, comparison, outcomes, and study design (PICOS) worksheet.

Parameters	Inclusion Criteria	Exclusion Criteria
**Participants**	Studies in humans Studies including COPD patients	In vitro and in vivo studies Participants with other malignancies
**Interventions**	Assessment of DNA methylation, Histone modification, and ncRNA expression	Others
**Comparisons**	Control Group	Others
**Outcomes**	(1) to provide unbiased and exhaustive overview of the current knowledge on the epigenetic modification associated COPD; (2) to summarize the epigenetic modifications translated into clinical therapeutic interventions and biomarkers for COPD.	Others
**Study Design**	Original studies in English	Review, Scoping Review, Narrative Review, Systematic Review, Meta-Analysis, Editorial, Book, Case Report, Conference Review, and Conference Paper

**Table 4 ijms-26-02571-t004:** DNA methylation in COPD.

Study	Country	Number of Participants	Type of Sample	Gene Affected	Epigenetic Alteration	Activity in COPD	Role of Epigenetic Mechanisms
Kachroo P, et al., 2021 [14]	Boston (USA)	N = 78 fetal N = 160 adult COPD	Lung tissue	Transcription factors, oxido-reductase, VEGFA-VEGFR2	Hyper-/hypo-methylation	Air flow limitation, inflammation activation, lung remodeling	Fetal origin of COPD
Kachroo P, et al., 2020 [15]	Boston (USA)	N = 78 fetal N = 160 adult COPD	Lung tissue	Co-methylation: Wnt, Pi3K/AKT, MAPK, Hippo	DNA methylation imbalance	Low lung function	Fetal origin of COPD
Schwartz U, et al., 2023 [16]	Heidelberg and Munich (Germany) Huston (USA)	N = 3 control N = 3 COPD I N = 5 COPD II-IV	Parenchymal fibroblasts (lung tissue)	3 cluster of genes involved in cell proliferation, DNA repair and extracellular matrix organization	Hyper-/hypo-methylation	Low lung function	Kinetics of DNA methylation in COPD
Strom JE, et al., 2022 [17]	Northern Sweden	N = 15 control N = 18 COPD	Macrophage from Broncho alveolar lavage (BAL)	DMPs co-localized with COPD-associated SNPs	DNA methylation imbalance	---	Pathophysiology of COPD
Cordero AIH, et al., 2022 [10]	Vancouver (Canada)	N = 27 control N = 15 COPD	Small airway epithelial brushings and buffy coat blood	DNAmGrimAge	DNA methylation imbalance	Biomarker for assessing accelerated aging in the airways of individuals with COPD	Biomarker
Morrow JD, et al., 2020 [18]	Boston (USA)	N = 336 control N = 331 COPD	Blood samples	Pi3KCD cg03971555 cg12033075	Hyper methylation	Predictive biomarker	Biomarker
Zhang Z, et al., 2021 [19]	Wuxi (China)	N = 18/17 control N = 8/16 COPD	Lung tissue/bronchoscopies (bronco epithelial cells)	Nfr2	Hyper methylation	Increased oxidative stress and cell death	Therapeutic target
Chen Q, et al., 2022 [20]	Groningen (The Netherlands)	N = 966/8 control N = 595/14 COPD	whole blood/airway epithelial cells	AHRR cg05575921 cg21161138	Hypo methylation	Airway epithelial cell proliferation, dysregulate mitochondrial function, and reduce apoptotic processes	Therapeutic target

**Table 5 ijms-26-02571-t005:** Histone modification in COPD.

**Study**	**Country**	**Number of Participants**	**Type of Sample**	**Gene Affected**	**Epigenetic Alteration**	**Activity in COPD**	**Role of Epigenetic Mechanisms**
Günes GG, et al., 2022 [21]	Ghent (Belgium)	N = 40 control N = 111 COPD	Monocytes/lung tissue	PRMT7	Histone methylation	Chronic inflammation	Pathophysiology of COPD/Therapeutic target

**Table 6 ijms-26-02571-t006:** circular RNA in COPD.

Study	Country	Number of Participants	Type of Sample	Gene Affected	Epigenetic Alteration	Activity in COPD	Role of Epigenetic Mechanisms
Duan R, et al., 2020 [22]	Beijing (China)	N = 21 control N = 21 COPD	Peripheral blood mononuclear cells	Gene involved in natural killer T cell activation and T-helper cell differentiation	Differential expression in COPD compared to control	Immune balance alteration	Pathophysiology of COPD/Therapeutic target
Xie T, et al., 2024 [23]	Hainan (China)	N = 5 control N = 10 (acute and stable) COPD	Peripheral blood mononuclear cells	Caspase 1, IL-18, IL-1β	Upregulation hsa-circ_0008833-57aa	Pyroptosis	Pathophysiology of COPD
Liu P, et al., 2022 [24]	Anhui (China)	N = 3 control N = 3 COPD	Blood samples	miR-1273h-3p; miR-411-5p; miR-122-5p; miR-615-5p; miR-519d-3p; miR-485-3p; miR-3646; miR-4714-5p; miR-203b-5p; miR-193a-5p; miR-1261; miR-4690-5p; miR-939-5p; miR-9-5p; miR-2113; miR-7977	Upregulation circFCHO2; circMBOAT2, circPTPN22; circTBC1D22A; circACADM; circCKAP5	----	Pathophysiology of COPD
Zhang C, et al., 2022 [25]	Jiangsu (China)	N = 17 COPD non-smoker N = 23 COPD smoker	Lung tissue	miR-24/PHPPL2 axis	Downregulation Circ_0006892	Inflammatory injury	Pathophysiology of COPD
Wang Z, et al., 2021 [26]	Hebei (China)	N = 27 control N = 21COPD	Lung tissue	miR-145-5p/BRD4 axis	Upregulation Circ_ANKRDII	Inflammation, apoptosis and oxidative stress	Pathophysiology of COPD
Tang S, et al., 2023 [27]	Hefei (China)	N = 30 control N = 30 COPD	Plasma samples	----	Differential expression circ_0008882; circ_00089763; circ_00062683; circ_00077607	Immune balance alteration	Biomarkers
Shen X, et al., 2024 [28]	Jiangsu (China)	N = 29 control N = 41 COPD	Peripheral blood mononuclear cells	----	Differential expression circ_0049875 and circ_0042590	Acute exacerbation of COPD	Biomarkers

**Table 7 ijms-26-02571-t007:** Long non-coding RNA in COPD.

Study	Country	Number of Participants	Type of Sample	Gene Affected	Epigenetic Alteration	Activity in COPD	Role of Epigenetic Mechanisms
Wu S, et al., 2020 [29]	Taipei (Taiwan)	N = 35 control N = 64 COPD	Peripheral blood mononuclear cells	IL-8, VCAM1, E-SEL	Downregulation lncRNA-IL7R	Inflammatory processes	Pathophysiology of COPD
Zhou AY, et al., 2020 [30]	Xiangya (China)	N = 3 control N = 7COPD	Lung tissue	Notch1	Downregulation lncRNA- HOXA-AS2	Cell viability and Inflammatory processes	Pathophysiology of COPD
Wang Y, et al., 2020 [31]	Wuhan (China)	N = 80 control N = 80 stable COPD N = 80 AECOPD	Peripheral blood mononuclear cells	Mir-146a/TNFα, IL-6, IL-8, IL-1β, IL-17	Upregulation LncRNA-PVT1	Inflammatory processes	Prognostic biomarker
Liu S, et al., 2020 [32]	Wuhan (China)	N = 120control N = 120 stable COPD N = 120 AECOPD	Blood samples	miR-125b, miR-133, miR-146a, miR-203/TNFα, IL-6, IL-8, IL-1β, IL-17, IL-23	Upregulation LncRNA-MALAT1	Inflammatory processes	Prognostic biomarker
Liu P, et al., 2021 [33]	Shanghai (China)	N = 90 control N = 50 COPD	Blood samples	Mir-18a-5p/TNFα, IL-6, IL-8, IL-1β	Downregulation lncRNA-CASC2	Inflammatory processes	Diagnostic biomarker
Zhao S, et al., 2021 [34]	Jiangsu (China)	N = 150 control N = 70 COPD	Blood samples	Mir-181-5p/Wnt/b-catenin axis	Upregulation LncRNA-LUCAT1	Apoptotic/Inflammatory processes	Biomarker/therapeutic target
Dai Z, et al., 2022 [35]	Xiangya (China)	N = 8 control N = 5COPD	Lung tissue	Bcl-2	Upregulation lncRNA-HOTAIR	Apoptotic processes	Therapeutic target
Zong D, et al., 2022 [36]	Xiangya (China)	N = 10 control N = 10 COPD	Lung tissue	Mir-152-3p/ERK	Upregulation lncRNA-CCT1	Inflammatory processes	Therapeutic target

**Table 8 ijms-26-02571-t008:** MicroRNA in COPD.

Study	Country	Number of Participants	Type of Sample	Gene Affected	Epigenetic Alteration	Activity in COPD	Role of Epigenetic Mechanisms
Wang L, et al., 2022 [37]	Xiangya (China)	N = 12 control N = 12 COPD	Peripheral blood mononuclear cells	IL-8 signaling; ------ iCOS-iCOSL signaling	Aberrant expression: miR-4453; miR-4736; miR-3118; miR-6967-5p; miR-132-3p; miR-96-5p; miR-4497 ------ miR-16-5p; miR-1964-5p; miR-29b-3p; miR-2355-3p; miR-18a-5p; miR-1234-3p; miR-148-3p; miR-21-5p; miR-1184; miR-140-5p; miR-19b-3p; miR-223-3p; miR-1246; miR-130a-3p	Inflammatory processes	Pathophysiology of COPD
Hu J, et al., 2022 [38]	Wuhan (China)	N = 3/9 control N = 3/9 COPD	BALF/ blood samples	MAPK, RAS, FOXO	miR-129-5p; miR-3529-3p; miR-365b-3p; miR-6503-5p; miR-26-3p; miR-34b-5p; miR-4748; miR-491-5p; miR-158-3p	Oxidative/inflammatory process	Pathophysiology of COPD
Zhang J, et al., 2020 [39]	Xuzhou (China)	N = 14/75 control N = 36/53 COPD	Alveolar macrophages/Peripheral blood mononuclear cells	HAT1/TNFα-IL-6-IL-8	Upregulation miR-486-5p	Inflammatory processes	Pathophysiology of COPD
Yang H, et al., 2021 [40]	Xuzhou (China)	N = 27 control N = 21COPD	Lung tissue	CDKN1B	Downregulation miR-221-3p	Apoptotic and Inflammatory processes	Pathophysiology of COPD
Chang C, et al., 2024 [41]	Beijing (China)	N = 19 control N = 13 COPD	Lung tissue	RAGE	Downregulation miR-23a-5p	Oxidative/inflammatory process	Pathophysiology of COPD
Kim R, et al., 2021 [42]	Newcastle, New South Wales (Australia)	N = 5 control N = 10 COPD	Lung tissue	SATB1/S100A9/NF-kB	Upregulation miR-21	Inflammatory processes	Pathophysiology of COPD
De Smet E, et al., 2020 [43]	Ghent (Belgium)	N= 44 control N= 48 COPD	Lung tissue	MMP12, ADAM19	Upregulation miR-155	Inflammatory processes	Pathophysiology of COPD
Zhu Y, et al., 2022 [44]	Brigham (USA)	N= 17control N = 8COPD	Alveolar macrophages	LDLR	Downregulation miR-103a	Oxidative/inflammatory process	Pathophysiology of COPD
Yang Z, et al., 2023 [45]	Suzhou (China)	N = 14 control N = 14 COPD	Lung tissue	NOS1	Upregulation miR-4640-5p	Pulmonary hypertension	Pathophysiology of COPD
Tasena H, et al., 2022 [46]	Groningen (The Netherlands)	N = 6 control N = 3 COPD	Lung tissue	MICS5AC, COL4A1, COL5A1	Upregulation miR-708-5p, let-7a-5p, miR-31-5p, miR-146a-5p	Epithelial differentiation, chronic mucus hypersecretion (CMH)	Pathophysiology of COPD
Singh P, et al., 2024 [47]	Birmingham (USA)	N = 9 control N = 13 COPD	Lung tissue	ANO1	Downregulation miR-381	Mucus production and secretion	Pathophysiology/therapeutic target
Zheng L, et al., 2021 [48]	Hefei (China)	N = 400 control N = 400 COPD N = 50 control + COPD	Blood samples/ Lung tissue	E-cadherin, α-SMA, Vimentin, N-cadherin	Downregulation miR-30	Epithelial-mesenchymal transition	Pathophysiology of COPD
Shi X, et al., 2022 [49]	Qinghai (China)	N = 40 control N = 40 COPD	Blood samples	------	Upregulation miR-486-5p; miR-106b-5p	Hypoxia/Pulmonary Hypertension	Biomarkers
Shen Y, et al., 2021 [50]	Nanjing (China)	N = 77 control N = 155 COPD	Blood samples	TNFα, IL-6, IL-8, IL-1β	Upregulation miR-221-3p; miR-92a-3p	Inflammatory processes	Biomarkers
Zhang L, et al., 2022 [51]	Guizhou (China)	N = 6 control N = 6 COPD	Blood samples	SLC17A9	Downregulation miR-548ar-3p	------	Biomarkers
Nadi E, et al., 2022 [52]	Hamadan (Iran)	N = 60 control N = 60 COPD	Blood samples	------	Downregulation miR-146a; miR-216	Oxidative/inflammatory process	Biomarkers
Ding Y, et al., 2023 [53]	Hefei (China)	N = 26 control N = 59 COPD	Blood samples	------	Downregulation miR-150-5p	Inflammatory processes	Biomarkers
Zhang X, et al., 2022 [54]	ChongQing (China)	N = 33 control N = 36 COPD	Blood samples	------	Downregulation miR-423-5p	------	Biomarkers
Tao S, et al., 2024 [55]	Xiangya (China)	N = 23 control N = 240 COPD	Blood samples	------ ------ PIK3R2	Downregulation miR-1290; miR-1246 ------ Upregulation miR-4433a-5p	------ ------ Apoptotic and Inflammatory processes	Biomarkers
Cazola-Rivero S, et al., 2020 [56]	Tenerife (Spain)	N = 13 control N = 24 COPD	Blood samples	MAPK, chemokines, Wnt	Downregulation miR-1246	Emphysema development	Biomarkers
Wang C, et al., 2022 [57]	Jiaozuo (China)	N = 70 control N = 140 COPD	Blood samples	TNFα, IL-1β, IL-6	Upregulation miR-126	Inflammatory processes	Biomarkers
Huang H, et al., 2020 [58]	Kunshan (China)	N = 80 control N = 160 COPD	Blood samples	------	Upregulation miR-210	Pulmonary hypertension	Biomarkers
Burke H, et al., 2022 [59]	Southampton (United Kingdom)	N = 20 control N = 24 COPD	BALF/Extra vesicles	------	Downregulation miR-338-3p; miR-204-5p ------ Upregulation miR-223-3p; miR-182-5p; miR-2110	Inflammatory patterns	Biomarkers
Wang F, et al., 2023 [60]	Peking (China)	N = 42 control N = 111COPD	Blood samples/Exosome	------	Upregulation miR-1258	Inflammatory processes	Biomarkers
Shen HF, et al., 2021 [61]	Binzhou (China)	N = 20 control N = 20 COPD	Blood samples	ARHGEF12, BCAT1	Upregulation miR-196-5p Downregulation miR-361-5p	Epithelial hyperplasia	Therapeutic target

**Table 9 ijms-26-02571-t009:** Competing endogenous RNA in COPD.

Study	Country	Number of Participants	Type of Sample	Gene Affected	Epigenetic Alteration	Activity in COPD	Role of Epigenetic Mechanisms
Li B, et al., 2023 [11]	Ningxia (China)	N = 6 control N = 14 COPD	Lung tissue	18 hub genes	ceRNA aberrant expression	Immune cells infiltration/differentiation; cell proliferation	Pathophysiology of COPD
Feng X, et al., 2023 [9]	Ningxia (China)	N = 6 control N = 7COPD	Lung tissue	TNFα/NF-kb; IL-6/JAK/STAT3	ceRNA aberrant expression	Inflammatory processes	Pathophysiology of COPD

## Data Availability

Registration to Open Science Framework platform (available online at https://osf.io/4ja25/; accessed on 31 January 2025) and Appendix A.

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
