# Peer review of "Recent Evidences of Epigenetic Alterations in Chronic Obstructive Pulmonary Disease (COPD): A Systematic Review"

_ijms, 2025, doi:10.3390/ijms26062571_

Round 1
Reviewer 1 Report
Comments and Suggestions for Authors
This is a very well written review concerning COPD. I would like to suggest an integrative figure or graphical abstract. A visual summary will help the general readers.
- This is a systematic and very complete review on Chronic Obstructive Pulmonary Disease. It is not an original research.
- As far as I know there is no similar systematic review on this important health issue. I think tat there is no a compared and recent other material recently published.
- The authors follow the "Preferred Reporting Items for Systematic Reviews and Meta-Analysis". They rigorously follow the methodology described by Moher et al. in PLoS Medicine (2009) and Page et al. in Syst. Rev (2021).
- This is not the case, because the paper is a Review. However they present clearly the general conclusions.
- The references appropriate.
- I would like to insist that a graphical abstract could help the readers.
Author Response
This is a very well written review concerning COPD. I would like to suggest an integrative figure or graphical abstract. A visual summary will help the general readers.
This is a systematic and very complete review on Chronic Obstructive Pulmonary Disease. It is not an original research.
As far as I know there is no similar systematic review on this important health issue. I think that there is no a compared and recent other material recently published.
The authors follow the "Preferred Reporting Items for Systematic Reviews and Meta-Analysis". They rigorously follow the methodology described by Moher et al. in PLoS Medicine (2009) and Page et al. in Syst. Rev (2021).
This is not the case, because the paper is a Review. However, they present clearly the general conclusions.
The references appropriate.
I would like to insist that a graphical abstract could help the readers.
We thank the Reviewer for this comment. A graphical abstract was added in the new version of manuscript. Moreover, additional figures (Figure 1 and 2) have been added in the new version of the manuscript to graphically summarized physiological and molecular features of COPD described in the introduction section.
Reviewer 2 Report
Comments and Suggestions for Authors
Various epigenetic pathways may affect the expression of multiple inflammatory genes in COPD patients. Yet, limited data exist about the epigenetic mechanisms involved in COPD. The clarification of these mechanisms could lead to novel therapeutic targets in patients with COPD. In this context, the present paper by Ragusa et al is quite welcome as it systematically summarizes previous knowledge on this subject. In general, the review is concise, well-written and interesting. I have only one major comment. Why did the authors limit their search to a rather arbitrary time frame (from 01- 01-2020 to the date of the search (09-05-2024)? Although specific guidelines on the limits of search dates are lacking, it is common practice to limit the search to the most recent 5 years or the most recent 10 or 15 years. Moreover, isn’t this restriction in discordance with the chosen general title of the present work?
Author Response
Various epigenetic pathways may affect the expression of multiple inflammatory genes in COPD patients. Yet, limited data exist about the epigenetic mechanisms involved in COPD. The clarification of these mechanisms could lead to novel therapeutic targets in patients with COPD. In this context, the present paper by Ragusa et al is quite welcome as it systematically summarizes previous knowledge on this subject. In general, the review is concise, well-written and interesting. I have only one major comment.
Why did the authors limit their search to a rather arbitrary time frame (from 01- 01-2020 to the date of the search (09-05-2024)? Although specific guidelines on the limits of search dates are lacking, it is common practice to limit the search to the most recent 5 years or the most recent 10 or 15 years.
We thank the Reviewer for appreciating our systematic review. We used the selected time frame to focus on the most recent discoveries related to epigenetic mechanisms and COPD and to avoid the massive inclusion of papers without dated analysis techniques. This explanation was reported in the manuscript in the Materials and Methods section.
- Materials and Methods, Paragraph 2.1.1. Systematic search phases, Pag. 4, Lines 121-124.
“We limited our search to articles, English, published from 01-01-2020 to the date of the search (09-05-2024) to include only the most recent papers in our systematic review and to avoid the massive inclusion of papers without dated analysis techniques.”
Moreover, isn’t this restriction in discordance with the chosen general title of the present work?
We thank the Reviewer for this comment and, accordingly, the Title was modified in the new version of the manuscript.
Title
“Recent evidences of epigenetic alterations in chronic obstructive pulmonary disease (COPD): A systematic review”
Reviewer 3 Report
Comments and Suggestions for Authors
This review manuscript submitted by Rosetta Ragusa et al. presents “ Epigenetic mechanisms in Chronic Obstructive Pulmonary Disease (COPD): A systematic review”
The manuscript is well-structured and provides valuable insights into the epigenetic mechanisms of COPD, but too much wording only makes it hard to follow the story. so it was difficult to catch up with the author's suggestions point. So, authors need to add more visual aids, such as diagrams and flowcharts, which can help illustrate complex concepts and improve understanding The current manuscript needs to improve the presentation and the sentence for the general reader.
Regarding “3.5 Histone modification”; the authors review only one publication, so should more review not only a single publication. Give more detailed information to the reader.
Figure 1: The flowcharts are unclear, particularly in column 3. There is no apparent relationship between the boxes. Consider widening the flowcharts and increasing the font size for better readability.
Table 1: Clarify the meaning of the “*” in each keyword.
Table 2: The table requires better formatting to enhance readability. The purpose of “((((((((((((((((Biomarker” is unclear.
Table 3: Correct the typo “Hystone” to “Histone.”
There are several typos regarding TNF-alpha and IL-1beta in the main text and tables.
For the supplementary data, provide exact titles for each table (e.g., Table S1, Table S2, etc.).
Comments on the Quality of English LanguageN/A
Author Response
This review manuscript submitted by Rosetta Ragusa et al. presents “Epigenetic mechanisms in Chronic Obstructive Pulmonary Disease (COPD): A systematic review”.
The manuscript is well-structured and provides valuable insights into the epigenetic mechanisms of COPD, but too much wording only makes it hard to follow the story. So, it was difficult to catch up with the author's suggestions point. So, authors need to add more visual aids, such as diagrams and flowcharts, which can help illustrate complex concepts and improve understanding. The current manuscript needs to improve the presentation and the sentence for the general reader.
We thank the reviewer for this comment. A graphical abstract and two novel figures (Figure 1 and Figure 2) were added in the new version of manuscript. The manuscript was carefully checked and modified when necessary in order to remove grammar/syntax mistakes.
Regarding “3.5 Histone modification”; the authors review only one publication, so should more review not only a single publication. Give more detailed information to the reader.
We understand the reviewer's concern. Previous studies (before 2020) reported that histone acetylation is associated with COPD progression and only one publication was published during the time frame considered for our systematic review. A new sentence explaining this issue was added in the new version of the manuscript.
- Results, Paragraph 3.5. Histone modifications, Pag. 15, Lines 325-328.
“Histone modifications are epigenetic mechanisms involved in regulation of gene transcription, genomic instability, premature aging, inflammatory activation and DNA damage/repair (Figure 2) [24]. Several studies before 2020 had already suggested that Histone modifications are associated with COPD presence and progression [25].”
Figure 1: The flowcharts are unclear, particularly in column 3. There is no apparent relationship between the boxes. Consider widening the flowcharts and increasing the font size for better readability.
We thank the reviewer for pointing out that. We increased the font size to enhance the readability. Regarding the relationship between the boxes in column 3: Figure 1 is the standardized PRISMA flowchart used to summarize the various steps of the Systematic Reviews following the PRISMA method (Moher et al, 2021; Page at al., 2021), column 3 identifies the papers eliminated step by step. For example, in our case, we had 5414 papers identified through Pubmed and Scopus, before starting the screening of titles and abstracts we needed to eliminate duplicates between the two databases, which turned out to be 2320. Thus, we subtracted the duplicates from the initial pool and started from a base of 3094 papers for title and abstract screening. Then, the number present in the box in column 3, has to be subtracted from the number present in regard to column 2 in the same row and we get the number present in the box in column 2 in the next row.
Table 1: Clarify the meaning of the “*” in each keyword.
We thank the reviewer for raising this issue. The use of “*” is a common practice during keywords selection in a systematic review; it is used to allow search engines to include the designated word regardless of what is from the asterisk onward. For example, entering “biomarker*” will allow the system to search for both “biomarker” and “biomarkers,” so it is very useful for including even plural words without entering the same word twice.
Table 2: The table requires better formatting to enhance readability. The purpose of “((((((((((((((((Biomarker” is unclear.
We thank the reviewer for raising this issue. Table 2 called Search Strategy is a standard table that is used in systematic reviews with the PRISMA method, it is very useful because it allows reproducibility of the search. All the parentheses that you see are due to the fact that the one reported is the exact search string that is produced by PubMed during a search; and to strictly follow the PRISMA method these should be reported in full in the Search Strategy table.
Table 3: Correct the typo “Hystone” to “Histone.”
The mistake was corrected in the last version of manuscript
There are several typos regarding TNF-alpha and IL-1beta in the main text and tables.
We thank the reviewer for the comment and as suggested TNFα and IL-1b have been standardized in the new version of the manuscript.
For the supplementary data, provide exact titles for each table (e.g., Table S1, Table S2, etc.).
Thank you for asking that. The titles for each table are listed at the bottom of the paper in the section Supplementary Materials:
- Supplementary Table S1: PRISMA checklist;
- Supplementary Table S2: AXIS tool;
- Supplementary Table S3: PRISMA abstract checklist.
Round 2
Reviewer 2 Report
Comments and Suggestions for Authors
I think that the authors have adequately addressed my comments in the
revised version of the manuscript.